# HIV Viral Load Estimation Using Hematocrit Corrected Dried Blood Spot Results on a BioMerieux NucliSENS^®^ Platform

**DOI:** 10.3390/diagnostics9030086

**Published:** 2019-07-30

**Authors:** Charles Nyagupe, Hemant Deepak Shewade, Serge Ade, Collins Timire, Hannock Tweya, Norah Vere, Sandra Chipuka, Lucia Sisya, Hlanai Gumbo, Ezekiel Ditima, Sekesai Zinyowera

**Affiliations:** 1National TB and HIV/AIDS Programme, Ministry of Health and Child Care, Harare P O Box CY1122 Causeway, 00263, Zimbabwe; 2International Union against Tuberculosis and Lung Disease (The Union), 75006 Paris, France; 3The Union South-East Asia, New Delhi 110016, India; 4Karuna Trust, Bengaluru 560041, India; 5Faculty of Medicine, Université de Parakou, Parakou BP:123, Benin; 6The Union Zimbabwe Office, Harare 00263, Zimbabwe; 7Light-House Trust, Lilongwe 00265, Malawi

**Keywords:** viral load, dried blood spot, Antiretroviral therapy (ART) monitoring, human immunodeficiency virus, hematocrit

## Abstract

While reporting human immunodeficiency virus (HIV) viral load (VL) using dried blood spot (DBS) in the BioMerieux NucliSENS platform, application of the hematocrit correction factor has been suggested. In this cross-sectional study from the National Microbiology Reference Laboratory of Zimbabwe, we assessed whether hematocrit correction (individual and/or mean) in DBS results improved the correlation with plasma VL and prediction of VL non-suppression (≥1000 copies per ml in plasma). Of 517 specimens during August–December 2018, 65(12.6%) had non-suppressed plasma VL results. The hematocrit correction factor ranged from 1.3 to 2.0 with a mean of 1.6, standard deviation (SD: 1.5, 1.7). The intraclass correlation (ICC) for mean (0.859, 95% CI: 0.834, 0.880) and individual (0.809, 95% CI: 0.777, 0.837) hematocrit corrected DBS results were not significantly different. The uncorrected DBS results had a significantly lower ICC (0.640, 95% CI: 0.586, 0.688) when compared to corrected DBS results. There were no significant differences in validity, predictive values, and areas under the receiver operating characteristics curves for all three DBS results when predicting VL non-suppression. To conclude, hematocrit correction of DBS VL results improved agreement with the plasma results but did not improve prediction of VL non-suppression. The results were not significantly different for individual and mean corrected results.

## 1. Introduction

To end the human immunodeficiency virus (HIV) epidemic by 2030, there was a recommendation from the Joint United Nations Programme on HIV/AIDS (UNAIDS) in 2014 that 90% of persons living with HIV (PLHIV) who were on antiretroviral therapy (ART)should be virally suppressed by 2020 [1]. HIV viral load(VL)testing at 6 months and 12 months after ART, and every 12 months thereafter is recommended by World Health Organization (WHO) for monitoring treatment response [1,2,3]. Less than 1000 HIV Ribose Nucleic Acid(RNA) copies per ml of plasma (gold standard specimen for VL testing) are accepted as an indicator for VL suppression (VL < 1000 copies per mL). If the VL is ≥1000 copies per mL, enhanced adherence counseling with repeat VL testing after three months is recommended. This is followed by a switch to a second line ART regimen if the VL remains non-suppressed [2].

The introduction of routine VL testing also brought its challenges to health systems of poorly resourced low and middle income countries, especially in southern Africa where the burden of HIV is high [4]. These include handling of whole blood which contains plasma specimen [3]. To address this challenge, dried blood spot (DBS) was introduced [3,5]. Blood spots are collected on a DBS card. Once dried, they are not considered hazardous. They can be stored for up to three months and transported at an ambient temperature without the loss of the viral genetic material [5,6]. DBS specimens are also thought to under-estimate VL results as compared to plasma specimens. This may result in wrong clinical decisions, especially if the VL is around a threshold of a 1000 copies per mL [7]. To compensate for the fact that DBS contains whole blood (with approximately 50% red blood cells) instead of plasma, various VL testing platforms recommend a correction factor to make the results comparable with plasma results [7]. This is especially important among specimens from PLHIV who may have a high prevalence of anemia [4].

A systematic review in 2014 revealed that the sensitivity of DBS VL is 78–100% compared to plasma VL at a threshold of 1000 copies per mL; and it increased to 100% at a threshold of 5000 copies per mL [8]. Four out of 13 studies included in the review reported that DBS VL were corrected for hematocrit [8]. In 2015, another systematic review suggested that the performance of DBS specimens was found to have an acceptable sensitivity and specificity for identifying virological failure when compared to plasma for a cut-off of 1000 copies per mL for most commonly used technologies. Based on this and other modelling studies, the WHO 2016 consolidated guidelines recommend that if VL testing with DBS specimens can be performed with reasonable sensitivity and specificity (>85%) when compared to plasma then costs and outcomes are similar [2].

The NucliSENS Easy Q^®^ HIV-1 v 2.0 (henceforth referred to as NucliSENS) is a platform for nucleic acid sequence-based amplification assay (NASBA) and has a sensitivity of 84% and a specificity of 95% for a threshold of 1000 copies per mL [2,3]. To improve the sensitivity, further research and development has been recommended which includes applying better correction factors to account for hematocrit [7]. Previous studies on validity of DBS VL using NASBA either did not clarify whether a correction factor was used or whether a correction factor based on mean hematocrit among PLHIV was applied or did not compare with plasma VL (or any other standard) against a threshold of 1000 copies per mL [9].

The aim of this study was therefore to determine whether hematocrit correction in DBS results (individualized and/or mean) improved the VL estimation and prediction of VL non-suppression (≥1000 copies per mL) when compared to plasma results.

## 2. Materials and Methods

### 2.1. Study Design

This was a cross-sectional study using secondary data.

### 2.2. Setting

The study was carried out at the National Microbiology Reference Laboratory (NMRL), Harare, the capital city of Zimbabwe, a landlocked country of 10 provinces and 63 districts. Zimbabwe is located in southern Africa, with approximately 13.6million inhabitants [10]. The country has a high prevalence of HIV (≈14.6%) among adults aged 15–49 years [11].

#### 2.2.1. National VL Program and its Microbiological Reference Laboratory

Zimbabwe’s National VL program is an arm of the AIDS department under the AIDS and TB program which is within the Ministry of Health and Child Care. In 2017, VL coverage among those on ART was around 30%, as per unpublished program reports. The country has eight VL testing sites. Specimen collection and transportation is done in such a way that the specimens reach the nearest VL testing sites on time. The NMRL is one of the two sites which have the NucliSENS platform which runs DBS as well as plasma specimens. More than 75% of the VL specimens received at NMRL are DBS, received mainly from peripheral areas. The other platforms for VL available in the country are Abbott which also processes both DBS and plasma and Roche which only analyses plasma specimens.

#### 2.2.2. VL Testing Procedure

##### VL Specimen Collection Procedure

At Harare hospital adult opportunistic infections ART clinic, venous whole blood is collected in 5 mL ethylene-diamine tetra-acetic acid (EDTA) vacutainers tubes. For DBS specimens, pre-perforated cards are spotted with 50 µL of whole blood on each of the 5 spots and left to air dry overnight. The DBS are packaged in sachets together with desiccants and humidity indicators, before they are transported to the laboratory. The DBS specimen is stable at an ambient temperature for three or more months. At the laboratory, the remaining whole blood is spun in a centrifuge at 3500 rpm for 15 min to get the plasma specimen. The plasma is aliquoted and stored in cryo-tubes. The plasma is either processed immediately or stored at −30 °C for not more than 30 days. Since the Harare hospital adult OI/ART clinic and NMRL are on the same premises, specimens are sent by a human carrier as a part of integrated specimen transportation.

##### Processing

DBS and plasma specimens processing are done following the standard operating procedures for BioMerieux NucliSENS Easy Q HIV-1 v2.0 which were adopted from the manufacturer by NMRL. Processing consists of two main activities: extraction of the HIV-RNA and its amplification (see Box 1 for DBS and plasma processing) [12]. Laboratory scientists adhere to a daily maintenance schedule for the machine. Bi-annual and annual services for the extraction and the amplification machines respectively are provided by external engineers from the flow cytometry center. Data at NMRL are routinely recorded in an electronic VL register database.

Box 1Dried blood spot (DBS) and plasma HIV viral load testing process at the National Microbiology Reference Laboratory (NMRL), Harare, Zimbabwe (2018).
**DBS processing**
***Extraction***: A volume of 100 µL DBS specimen is required. Two pre-perforated DBS spots (volume of 50 µL each) are punched into a 2 mL lysis buffer tube using the back of a micropipette tip. They are incubated and mixed by rocking them for 30 min on a rocker. The lysis tubes are then centrifuged at 3500 revolutions per minutes (rpm) for one minute. The lysate consisting of 100 µL concentration of the sample in 2 mL lysis buffer is transferred using a filtered 1000 µL micropipette tip to specimen holding containers of the NucliSENS which are called disposables. Hundred microliters concentration of the specimen is set on the machine. The premix consisting of a calibrator and silica are added to the DBS lysate and the contents are loaded onto the machine for extraction of the viral RNA (this extraction process is referred as the Boom technique). The extraction process takes 40 min and yields 25 µL of eluate of nucleic acids. ***Amplification***: A starting volume of 15 µL is needed for both DBS and plasma samples. A volume of 15 µL of viral RNA eluate is harvested in PCR tubes for amplification purposes. Twenty microliters of the primer is added to the eluate. The mixture is incubated for 270 s. Five microliters of avian myeloblastosis virus, reverse transcriptase, T7 RNA polymerase and RNase H enzyme are added to this mixture. The mixture is then vortexed two to three times for two seconds and micro-centrifuged for two seconds. The next step is loading of the mixture onto the amplification machine. The amplification is isothermal and is carried out at 41 °C for one hour. The process is RNA specific and is called nucleic acid sequence based amplification (NASBA).
**Plasma processing**
At the NMRL, plasma is prepared from whole blood by spinning 5 mL EDTA tube with whole blood at 3500 rpm for 15 min. The plasma is aliquoted into cryotubes and is either tested immediately or stored at −30 °C for testing within 30 days. ***Extraction***: Like the DBS specimen, a starting volume of 100 µL is also needed. A volume of 100 µL of the plasma specimen is transferred into a disposable and loaded onto the machine which is pre-set at that volume. 2 mL of lysis buffer is dispensed from the machine and after 10 min of incubation, a pre-mixture of the calibrator and silica is added to the lysed plasma for extraction process to begin. Like DBS, the extraction process takes 40 min. ***Amplification***: The amplification process is the same as that in DBS and is completed in one hour.

### 2.3. Study Population

All PLHIV on ART from Harare hospital’s adult OIC/ART clinic who underwent both DBS and plasma VL testing on the NucliSENS platform at the NMRL, and who had a valid result for hematocrit between August and December 2018 were included.

### 2.4. Variables, Sources of Data, and Data Collection

Data were extracted from the electronic database at NMRL in Microsoft Excel (Microsoft, Redmond, WA, USA) format in January 2019. Demographic characteristics, DBS and plasma VL results (in copies per mL) were collected. For specimens with undetectable VL, a value of zero was assigned. For specimens with detectable VL below the limit of detection (<100 copies per mL), a random value between 1 and 99 was assigned. Hematocrit data were manually extracted from the adult OI individual files.Full blood count which includes hematocrit was performed using a 5-part auto analyzer (Sysmex XT-4000i, Sysmex Corporation, Chuo- ku, Kobe, Japan).

### 2.5. Data Analysis

Electronic data were cleaned and then imported into Stata v 12.1 (Stata Corp, College Station, TX, USA) for analysis. The correction factor was derived from the formula “100/(100-hematocrit)”. The DBS VL result was multiplied by, first the individual and then by the mean hematocrit correction factor to obtain two corrected DBS VL results. The mean hematocrit was determined from the study population. 

The intraclass correlations (ICC, 95% confidence interval (CI)) of the uncorrected and corrected (individual and mean hematocrit) DBS VL with plasma VL results were calculated. ICC using absolute agreement (two-way mixed, single measure) was calculated [13]. Depending on the ICC values and their 95% CI, correlation were classified as poor (<0.50), moderate (0.50–0.75), good (0.76–0.90) and excellent (>0.90) [13].

For a cut-off of 1000 copies per ml in plasma, the receiver operating characteristics (ROC) of the uncorrected and corrected (individual and mean hematocrit) DBS VL was described. Area under the ROC curve is an indicator of the accuracy of test compared to the plasma test. 

Finally, the sensitivity, specificity, positive and negative predictive values (with 95% CI) of uncorrected and corrected DBS results in predicting VL non-suppression in plasma were determined. 

### 2.6. Ethics Approval

The study was approved by the Medical Research Council of Zimbabwe (MRCZ No. E/220, dated 06 December 2018) and from The Union Ethics Advisory Group, Paris, France (EAG No. 51/18 dated 11 September 2018). As the study involved review of routinely collected secondary data, a waiver of informed consent was sought and approved by the ethics committees. 

## 3. Results

A total of 554 specimens were received. Of them, 517 (93.3%) had valid results for both plasma and DBS specimens and had a hematocrit result available and were therefore included in the study. Of the 517, the mean age (SD) was 42.2 (15.2%) years; 172 (33.3%) were males; 272 (52.6%) were females and sex was not recorded for 73 (14.1%).

The median plasma and DBS VL was 0 (IQR: 0, 0) and 24 (IQR: 0, 260), respectively. A total of 399 (77.1%) had undetectable plasma VL (10.3%), 53 had detectable plasma VL and 65 (12.6%) had non-suppressed plasma VL. A total of 235 (45.5%) had undetectable DBS VL (42.9%), 222 had detectable DBS VL and 60 (11.6%) had non-suppressed DBS VL results. Of 235 with undetectable DBS VL, 22 (9.4%) had detectable plasma VL. A total of 30 (5.8%) DBS specimens had VL between 500 and 999. Of these 30, nineteen had plasma VL < 1000.

The minimum, maximum and mean (SD) hematocrit was 23.7%, 50.9% and 37.9% (5.4%), respectively. The minimum, maximum and mean (SD) hematocrit correction factor was 1.3, 2.0 and 1.6 (0.1), respectively.

The ICCs between DBS (uncorrected, mean hematocrit corrected and individual hematocrit corrected) and plasma VL results are depicted in Table 1. The ICC along with the 95% CI was within 0.50 and 0.75 for uncorrected (moderate correlation) DBS while it was within 0.76 and 0.90 for both the corrected DBS results (good correlation). The ICC was significantly higher for corrected DBS results when compared to uncorrected DBS results (95% CIs of ICC not crossing each other), the ICC for mean and individual hematocrit corrected DBS results were not significantly different. For the subgroup with plasma VL ≥ 1000, the uncorrected DBS results had poor to good correlation, mean corrected DBS results had good to excellent correlation and individual corrected DBS results had moderate to good correlation with plasma results. 

The validity and predictive values of DBS HIV-RNA in predicting VL non-suppression in plasma results is depicted in Table 2. High false positivity (>25%) and low false negativity (<5%) was observed with or without hematocrit correction in DBS results. With hematocrit corrections in DBS, the sensitivity improved from 68% to 80% (95% CIs crossing each other). The validity and predictive values were similar for mean and individual hematocrit corrected DBS results. However, for all three DBS results, the areas under ROC curves were similar (Figure 1).

## 4. Discussion

To our knowledge, this is one of the few studies among PLHIV on ART that attempted to include an individual and mean corrected hematocrit factor for DBS HIV VL results on a NucliSENS platform and evaluated its influence on estimating plasma HIV VL results and predicting viral non-suppression. Both mean and individual hematocrit correction of DBS VL results improved the correlation with plasma VL results and there was no significant difference between individual and mean corrected DBS VL results. DBS results overall had a high false positivity in predicting VL non-suppression for a threshold of 1000 copies per mL. However, hematocrit correction did not improve prediction of VL non-suppression.

### 4.1. Limitations

There were two limitations. First, majority of the patients were virally suppressed with less than 100 copies per ml or targets not detected (TND) and 30 DBS results were between 500 and 999 where correction would have possibly altered the clinical interpretation of viral suppression. This was possibly the reason for no significant difference in the results of uncorrected and corrected DBS. However, this represents the real-life scenario and reflects the reality on the ground. Second, the study was carried out in one site, in a defined period between August and December 2018. The study has to be repeated in other settings and conducted over a longer period of time to assess for consistency of findings and inform decision making (for example whether the mean hematocrit used during August–December 2018 and January–June2019 yields similar results). 

### 4.2. Key Findings and Implications

To predict viral non-suppression using a plasma VL threshold of 1000 copies/mL, the sensitivity of uncorrected DBS (67%) results was lower than reported by the WHO 2014 technical guideline (84%). However, on hematocrit correction, the sensitivity was similar to 84%. The specificity was however comparable [3]. A recent study from Belgium reported a sensitivity of 77% and a specificity of 93% on a NucliSENS platform after using a correction factor of 2.29 [7]. Evans C et al. have described that correlations between DBS and plasma drug measurements are weaker in ICU patients than in healthy volunteers [14]. Therefore, it might be important to take patients’ physical conditions into consideration.

Previously, variations between DBS and plasma VL results have been reported. For instance, in one study from Spain, Garido et al. reported that approximately 16% of DBS specimens with undetectable viremia were actually found to be positive with plasma results [15]. In another study, DBS quantification was on average 0.36 log10 lower, as compared to plasma VL [16]. In another one, VL values with DBS were also lower than in plasma with a mean difference of 0.32 log (0.22) [17]. For all these studies, DBS was not corrected with hematocrit, suggesting the role of hematocrit correction in the differences that were found with plasma results. Our findings confirm that on addition of a hematocrit correction factor, the correlation with plasma VL improved. The type of correction could be based on either an individual or a mean hematocrit corrected factor, depending on the availability of resources. However, among those with plasma VL < 1000, we are not able to confidently recommend the use of DBS to assess relative change of VL over time (say from 400 to 800) which is also an early indicator of treatment failure [18,19]. This was due to negative average covariance in the subgroup, violating reliability model assumptions.

The CIs of the sensitivity, specificity, positive, and negative predictive values before and after correction were crossing each other. We did not find any benefit in adding hematocrit correction in predicting VL non-suppression. This could be because only a few PLHIV had a VL between 500 and 999.This suggests that when an absolute cut-off (VL ≥ 1000 copies per mL) is used for decisions related to possible treatment failure, it does not matter whether one uses corrected or uncorrected DBS. However, these findings may be confirmed in studies from other settings where a significant proportion of patients has their uncorrected DBS VL results between 500 and 999.

The low positive predictive value (high false positivity) of DBS could reflect low prevalence of VL non-suppression in the sample (PLHIV who were on ART). False negativity was very low. Given these results, though VL suppression based on DBS (irrespective of correction) can be confidently used to continue the ongoing ART regimen, VL non-suppression results based on DBS (especially after enhanced adherence counseling) should be preferably confirmed with plasma VL test.

To conclude, hematocrit correction of DBS HIV VL results on a NucliSENS platform improved the correlation with plasma VL results. No difference was found between individual and mean hematocrit corrected DBS. Among patients with plasma VL < 1000, the correlation of DBS was poor irrespective of hematocrit correction. Future research must be conducted in other routine settings and over a longer time period to ascertain the practicality of hematocrit correction, especially for prediction of viral non-suppression. For now, the program may safely act on a suppressed VL result using DBS on a NucliSENS platform. However, the program must be cautious in the interpretation of non-suppressed VL result using DBS (seen in approximately one in ten PLHIV on ART who undergo follow up VL testing).

## Figures and Tables

**Figure 1 diagnostics-09-00086-f001:**
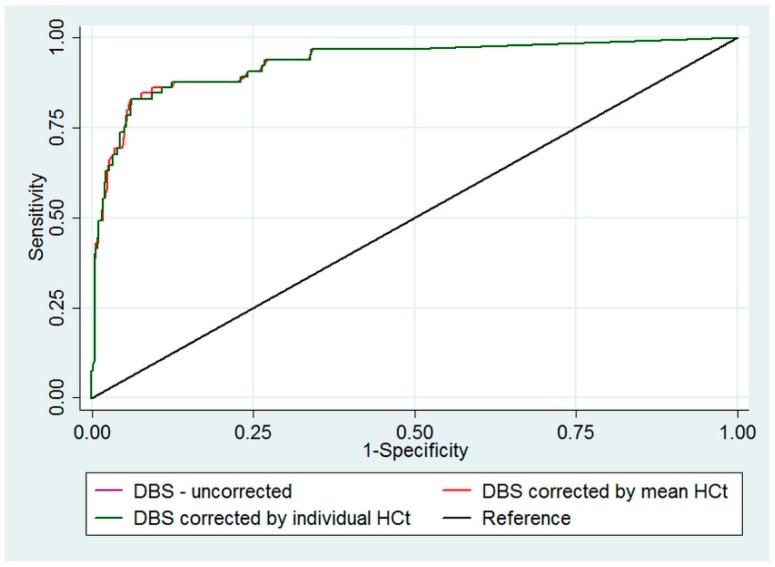
ROC curve*for uncorrected, mean hematocrit corrected and individual hematocrit corrected DBS HIV-RNA results in predicting VL non-suppression (≥1000 copies per mL) as measured in plasma, Zimbabwe National Microbiology Reference Laboratory, August–December 2018 (*n* = 517). ROC: receiver operating characteristic, DBS: Dried blood Spot; VL: viral load. * Area under curve for uncorrected DBS—0.933; mean hematocrit corrected DBS—0.933; individual hematocrit corrected DBS—0.932. The ROC curve for uncorrected DBS (purple line) is not seen as it is hidden below and overlapped by the ROC curve for mean corrected DBS (red line), their area under the curve are the equal.

**Table 1 diagnostics-09-00086-t001:** Intraclass correlation between plasma and DBS (uncorrected, mean hematocrit corrected and individual hematocrit corrected) HIV-RNA results, Zimbabwe National Microbiology Reference Laboratory, August–December 2018 (*n* = 517).

Intraclass Correlation (ICC) of Plasma with	ICC *	(95% CI)	*p*-Value
**All specimen (*n* = 517)**			
Uncorrected DBS	0.640	(0.586, 0.688)	<0.001
Mean hematocrit corrected DBS	0.859	(0.835, 0.809)	<0.001
Individual hematocrit corrected DBS	0.809	(0.777, 0.837)	<0.001
**Subgroup-plasma result ≥ 1000 (*n* = 65)**			
Uncorrected DBS	0.635	(0.462, 0.761)	<0.001
Mean hematocrit corrected DBS	0.857	(0.769, 0.912)	<0.001
Individual hematocrit corrected DBS	0.805	(0.694, 0.878)	<0.001
**Subgroup-plasma result < 1000 (*n* = 452) ^**			
Uncorrected DBS	0.000 ^	(−0.091, 0.091)	0.500
Mean hematocrit corrected DBS	0.000 ^	(−0.091, 0.091)	0.500
Individual hematocrit corrected DBS	0.000 ^	(−0.091, 0.091)	0.500

DBS = Dried blood Spot; * ICC using absolute agreement (two-way mixed, single measure); ^ negative average covariance among items. This violates reliability model assumption.

**Table 2 diagnostics-09-00086-t002:** Validity and predictive values of uncorrected, mean and individual hematocrit corrected DBS HIV-RNA results in predicting VL non-suppression (≥1000 copies per mL) in comparison with plasma results, Zimbabwe National Microbiology Reference Laboratory, August–December 2018 (*n* = 517).

DBS HIV-RNA Result	Total	Plasma HIV-RNA			Sensitivity (95% CI)	Specificity (95% CI)	PPV (95% CI)	NPV (95% CI)
		≥1000	<1000	Total				
		65	452	517				
**Uncorrected DBS**	≥1000	44	16	60	67.7 (54.8–78.5)	96.5 (94.2–97.9)	73.3 (60.1–83.5)	95.4 (92.9–97.1)
	<1000	21	436	457				
**Mean hematocrit** **corrected DBS**	≥1000	52	26	78	80.0 (67.9–88.5)	94.3 (91.6–96.1)	66.7 (55.0–76.7)	97.0 (94.9–98.3)
	<1000	13	426	439				
**Individual hematocrit** **corrected DBS**	≥1000	52	27	79	80.0 (67.9–88.5)	94.0 (91.3–95.0)	65.8 (54.2–74.9)	97.0 (94.8–98.3)
	<1000	13	425	438				

DBS: Dried blood Spot; PPV: positive predictive value; NPV: negative predictive value; 95%CI = 95% Confidence interval; DBS = Dried blood Spot; VL: viral load.

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
