# Peer review of "HIV Viral Load Estimation Using Hematocrit Corrected Dried Blood Spot Results on a BioMerieux NucliSENS® Platform"

_diagnostics, 2019, doi:10.3390/diagnostics9030086_

Reviewer 1 Report

In this manuscript, Nyagupe et al. evaluated hematocrit corrected DBS for HIV vRNA measurement. They found that DBS HIV vRNAs were better correlated with plasma vRNAs when corrected by either mean or individual hematocrit values. However, when DBS was used as a clinical test for detecting HIV VL 1000+, both methods of hematocrit correction resulted in increased sensitivities but also increased false positives. The results are interesting. But the given analysis might seem to be limited to fully describe the value of the study with such a large sample size. I suppose more data analysis could better characterize the given data and the suggested methods. Another concern is the low sensitivity observed with the uncorrected DBS method in the study. It should be clarified whether the study on hematocrit correction really makes sense in this setting.

1.      In Table 1, the authors show the correlation analysis results. The data on intraclass correlation (ICC) is comprehensible.

a)      The methods for the ICC analysis could be better described.

b)      The ICC analysis results could be further comprehensible by plotting the data with regression lines. This is because different factors can affect ICC correlation coefficient.

c)      I suppose the patients could be subgrouped for further correlation analysis. For example, isn’t it meaningful to set the following subgroups? (1) plasma vRNA(1000~) (2) plasma vRNA(500~999) (3) plasma vRNA(~499), where the preceding “plasma”s might be replaced with “uncorrected DBS”s. It might be meaningful to check in which group uncorrected DBS vRNA better correlated with plasma vRNA. This could be another indicator of the sensitivity of the DBS method and give insights to the Table 2 results.

2.      In Table 2, the sensitivity of the uncorrected DBS is described as 67.7 %. This is much lower than the sensitivity described in Introduction (84 %, page 2, line 70). The reason for the discrepancy should be discussed well to address any quality concerns of the given data.

a)      In page 5 line 178, the text describes high false positivity (>25 %) and low false negativity (<5 %) in all the DBS results. I am not quite sure of the uses of these terms. For example, I suppose false positivity is defined as (1 – specificity). Is it right?

3.      In Figure 1, the purple line showing uncorrected DBS results is not visible to me. The graph could be amended so that overlapping curves can be better distinguished. I wonder why the graph is not square; is there any scientific reason?

4.      In page 6 lines 210-211, the authors state that they had a few DBS results between 500 and 999. This is actually 30, quite a few, as described in page 4, line 165. The number can be shown here as well. The authors could also show whether the correction in the subgroup DBS(500-999) is actually effective by data rather than only by discussion.

5.      In page 6 lines 220-223, the sentences are not very comprehensible. The authors could better discuss the reason why hematocrit correction makes sense without misleading effects if the base sensitivity for the uncorrected DBS test is so low. The authors could compare their results with previously published results with or without hematocrit corrections.

6.      The text could be edited by scientists with native English proficiency.

Author Response

Thank you for the comments. Though the sensitivity improved from 68% to 80% (after correction), the 95% CIs were crossing each other (lines 192-97 of revised manuscript with track changes). Hence, we did not discuss this in later in the discussion section. However, in lines 242-47 of revised manuscript (discussion section) we have compared the sensitivity in our study and reported in a WHO 2014 update (68% vs 84%). We have also compared our findings with other studies.

Regarding whether the study on hematocrit correction really makes sense in this setting. Haematocrit correction of DBS VL results improved the correlation with plasma VL results but did not improve prediction of VL non-suppression. But we think that this finding (did not improve prediction of VL non-suppression) may be confirmed in studies from other settings where significant proportion of patients has their uncorrected DBS VL results between 500 and 999. (lines 268-274 of revised manuscript)

The results (ICC, ROC and validity) were not significantly different for individual and mean corrected DBS.

       1a. Thank you very much for this. We had missed describing this. In the methods section of revised manuscript, we have mentioned that ICC using absolute agreement (two way mixed, single measure) was calculated. In addition, we have described the criteria for interpretation ICC. We have also added a reference for this (reference number 13 in revised manuscript.

Please note that by mistake we had presented the ICC values (average measures) in the initial submission. ICC values (single measure) are more appropriate for our analysis. Keeping this point in mind, we have modified the results section including Table one. There has been no major change in interpretation of our findings as a result of this change.

For revisions, please refer to the lines 177-87 of revised manuscript with track changes.

       1b. We agree that difference factors can affect ICC correlation coefficient. But in this case, there were not many factors available with us. The plasma VL was the dependent variable (Y axis) and the DBS VL was the independent variable (X axis).

       1c. Thank you for this suggestion. We have done a sub-group analysis for plasma VL<1000 and="">/=1000. For plasma VL less than 1000 (n=452), the DBS had poor reliability irrespective of correction. For plasma VL ≥1000, the uncorrected DBS had moderate reliability and corrected DBS had good reliability (ICC results of this subgroup were similar to the overall ICC results). Please refer to lines 177-87 of revised manuscript with track changes.

We did not do a separate sub-group analysis for plasma VL 500-999 as there were only six patients in this sub-group.

       2.  Though the sensitivity improved from 68% to 80% (after correction), the 95% CIs were crossing each other. Hence, we did not discuss this in later in the discussion section. However, in line 242-47 of revised manuscript (discussion section) we have compared the sensitivity in our study and reported in a WHO 2014 update (68% vs 84%). We have also compared with other studies reporting similar sensitivity.

       2a. The false positives and False negatives were obtained by subtracting the PPVs  and NPVs from 100%, respectively. Thus 100-73.3% and (100-95.5%) gives us >25% and<5% respectively.

       3.   The lines overlap, making it difficult to distinguish them. We have added a footnote in Figure 1to explain this.

       4.   Subgroup analysis is done based plasma VL value, and not based on DBS VL. Plasma is the gold standard. As mentioned in line 167-69 of revised manuscript, 452 (out of 517) had a plasma VL<1000. Of them, only six had a plamsa VL between 500 and 999. Hence, when we did the subgroup analysis, we did it only for two subgroups: one with plasma VL<1000 and="" one="" with="" plasma="" vl="">/= 1000.

30 number of PLHIV on ART with DBS between 500 and 999 was mentioned in the results section because, it is these people in whom any haematocrit correction will change the DBS value with a chance that it may touch or cross 1000. It is the number of people with DBS between 500 and 999(if significant in number), who will contribute to the difference in AUC between uncorrected and corrected DBS. To avoid confusion, we have reorganized the results narrative (lines 167-71 of revised manuscript)

5.    The results were compared with findings from other studies. Please refer to lines 242-247 of revised manuscript. 

       6.    We have checked the whole document and corrected significant portions. We hope the English matches your expectations.

Reviewer 2 Report

This study evaluates the utility of adding a haematocrit correction factor of HIV viral load results from dried blood spots (DBS) in improving the agreement with HIV viral load from plasma and prediction of viral load non-suppression, relative to uncorrected DBS viral load.

The manuscript is well-written, the justification is presented clearly, and the methodology is sound. A few comments are provided to help further clarify the results:

1. The description of the sample processing is somewhat unclear (eg "Hundred microliters of the specimen is set on the machine"). It would be useful to provide the volumes of starting sample, nucleic acid elution volume, and volume used for amplification, for both DBS and plasma.

2. The numbering in the Materials and Methods skips from section 2.2 to section 2.4

3. This sentence is unclear: "For specimen with detectable VL below the limit of detection (<100 copies per ml), a random value between 1 and 99 was assigned". Is 100 cp/mL the limit of quantification, rather than the limit of detection? If so, what is the limit of detection? It seems that assigning a random value between 1 and 9 cp/mL may not be valid, if the sample processed is only 0.1 mL.

4. It would be helpful to list the number of specimens with undetectable VL versus detectable<100 cp/mL, particularly since it appears that the median and IQR for plasma was undetectable, whereas the median for DBS was <100 cp/mL with IQR up to 260 cp/mL. This should not impact the main focus of the manuscript, which is on VL non-suppression, but could inform whether DBS was more sensitive overall in detecting HIV relative to plasma.

5. In Figure 1, the purple curve corresponding to DBS uncorrected is not visible.

6. The data from this study show that haematocrit correction of DBS VL improves the sensitivity in predicting VL non-suppression. However, regardless of hematocrit correction, DBS VL is plagued by poor positive predictive value in this study, ie the false positive rate is high. Could the authors comment on the consequences of potentially unnecessary clinical decisions for viral non-suppression based on DBS VL? Or would there be a reflex to plasma VL for confirmation? 

Author Response

1. On both DBS and plasma samples the starting volume is 100ul.For DBS ,two pre-perforated disc of 50ul each of dried whole blood are punched into 2ml of lysis buffer.Elution is achieved by rocking the tubes for 30 minutes on a rocker.The whole volume of 2ml buffer with a concentration of 100ul DBS sample is loaded on the machine in sample holding vessels called disposables.The machine is pre-set at a concentration of 100ul.Extraction process takes place in 40minutes,yielding a volume of 25ul of viral RNA eluate.The volume used for amplication is 15ul for both plasma and DBS.

For plasma specimen,a 5ml tube of EDTA whole blood,is separated by spinning it at 3500rpm(raves per minute) in a centrifuge.The plasma is harvested in cryotubes for immediate processing or stored at -300c for later processing within 30 days.The start volume for a plasma sample is also 100ul.There is on-board lysis of the pre –loaded plasma sample where by 2ml lysis buffer is dispensed from the machine into the sample vessels.After 10minutes of incubation ,a premix of a calibrator and silica is added to the lysed sample and extraction process occurs within 40minutes.For both plasma and DBS ,extraction occurs by a physical process of separation of the negatively charged nucleic acid and positively charged silica,referred to as the Boom Technology.Like DBS sample,the extraction process yield 25ul eluate of viral RNA,and again a starting volume of 15ul is needed for amplification process to occur.

For both plasma and DBS eluates, amplification is isothermal and occurs at 410C by the process called Nucleic Acid Sequence Based Amplification(NASBA).The amplification process is real time and takes one hour.The results produced are manually entered into the National viral load database

We have revised these details in the box in the revised manuscript with track changes.

2.  Many thanks for this observation. Section 2.3 (study particpants was hidden behind the box). Hence, it appeared as if the section skips form 2.2 from 2.3. It now fine in the revised manuscript.

3. Many thanks for raising this. 100copies/mL is the limit of detection (LOD) of the machine. We define LOD here loosely as the lowest value of VL particles that can be accurately detected and quantified by the machine. The machine cannot give an absolute quantity of VL; it just gives a qualitative result (<100) for VL results of between 1-99, inclusive. For patients with VL of <100, we were not so sure about the actual value of VL result. We therefore randomly assigned values of between 1-99 for every patient with a VL of <100. We hope this is fine.

4. Many thanks for this. We have provided the details in the revised results narrative (lines 167-171). We are reproducing it below

“The median plasma and DBS VL was 0 (IQR: 0, 0) and 24 (IQR: 0, 260), respectively. A total of 399 (77.1%) had undetectable plasma VL (10.3%), 53 had detectable plasma VL and 65 (12.6%) had non-suppressed plasma VL results. A total of 235 (45.5%) had undetectable DBS VL (42.9%), 222 had detectable DBS VL and 60 (11.6%) had non-suppressed DBS VL results. A total of 30 (5.8%) DBS specimens had VL between 500 and 999”

5. The lines overlap, making it difficult to distinguish them. We have added a footnote underneath to explain this.

6. Risk benefit of false positive versus false negative. If PLHIV are incorrectly classified as VL non-suppressed, they undergo enhanced adherence counselling. In this case a false positive result is beneficial for the patient. Even after enhanced adherence counselling if VL>/=1000, we think a plasma VL may be considered before taking a call. We have incorporated these points in line 275-80 of revised manuscript (discussion section).

Greatest risk occurs when a PLHIV who was supposed to undergo enhanced adherence counselling do not go for the session (false negative is relatively very low).Bottom of Form

Round  2

Reviewer 1 Report

I understand that the authors have tried to improve the quality of the manuscript. But the revised manuscript might be partly more confusing than the initial manuscript. The conclusions in the last paragraph have been modified, while the authors have made no change in the abstract (and title) regarding this. The English writing has been partly modified, but I do not know whether this is highly effective. As a scientific paper, it might be desirable if the authors could better discuss the mechanisms for the lower sensitivity of the uncorrected DBS and the mechanisms for the improved sensitivity by correcting the data with hematocrit measurements.

1.      In page 5 lines 183-187, the authors discuss their subgroup analysis without data. I hope they could provide some data, as this analysis seems to have affected the conclusions of the study. The expressions “moderate reliability” and “good reliability” are not very clear.

2.      In Figure 2 legend, the authors describe “purple line is not seen as it is hidden below and overlapped by the other two lines”. This could be problematic in terms of scientific soundness.

a)      First, if the purple line is not visible in this paper, the figure should not indicate the purple line.

b)      Second, it is not clear which of the other two lines overlap the imaginary purple line. I wonder how difficult it would be for the authors to present the purple line separately.

3.      In page 7 lines 242-249, the sentences have been amended but possible reasons for the low sensitivity of uncorrected DBS in this study has not been discussed well. I think there could be a couple of problems.

a)      First, the paper could verify that the authors are aware of any technical and scientific reasons to lower the sensitivity of the DBS method.

b)      Second, this paper might demonstrate the improvement of low-quality DBS data (67 % by sensitivity) by hematocrit correction, although I am not sure whether the scientific reasons for the improvement could be connected with the scientific reasons to cause the low sensitivity. On the other hand, this paper might not be demonstrating further improvement of good-quality DBS data (~84 %) by hematocrit correction. This is partly discussed in page 8 lines 286-288, but I suppose the discussions could be further improved. For example, Evans C et al. (AAPS J. 2015, 17: 292-300) describes that correlations between DBS and plasma drug measurements could be weaker in ICU patients than in healthy volunteers. Therefore, it might be important to take patients’ physical conditions into consideration.

c)      I am not sure of the meaning of the correction factor of 2.29 in ref No. 7.

4.      I previously recommended English editing of the manuscript. I hope the authors could take it seriously.

5.      I think it is important to better clarify the subgroup DBS(500-999). How many of them (n=30) had plasma VL(1000-)? Is there any difference between the mean and individual hematocrit correction methods when analyzing the subgroup DBS(500-999)?

6.      If there is a correction factor recommended by the company (2.29), don’t the authors need to compare their hematocrit-based correction results with other correction methods including the method recommended by the company rather than with uncorrected DBS results? What do the authors think of the cons and pros of hematocrit-based correction methods against the correction factor recommended by the company?

7.      Could the authors clarify whether the “sensitivity" reported in the WHO guideline (84 %) was based on any correction such as the correction factors recommended by the companies or not? I think this is important and should be clarified.

Author Response

Top of Form

Thank you very much for the prompt second round of review. We really appreciate this.

Regarding not making changes in the conclusion section of abstract. The abstract section has a word limit of 200 words and it has to be standalone as well. Hence, we have decided to present the ‘key’ result here. The ‘key’ result is that hematocrit correction significantly improved correlation of DBS results with plasma VL result. However, there was no significant difference between mean or individual corrected DBS VL result. Hematocrit correction of DBS result did not improve the prediction of VL non-suppression.

We hope this is fine.

We have also made edits in the title. It now reads

“HIV viral load estimation using haematocrit corrected dried blood spot results on a BioMeriux NucliSENS platform”

Regarding discussing the corrections, sensitivity etc, we have done it in the introduction section. Hence, we did not feel the need to repeat the same in the discussion section. We hope this is fine.

Please refer to lines of revised manuscript with track changes (introduction section)

Line 52-64 reproduced below

“The introduction of routine VL testing also brought its challenges to health systems of poorly resourced low and middle income countries, especially in Southern Africa where HIV burden is high[4]. These include handling of the whole blood which contains plasma specimen[3].To address this challenge, dried blood spot (DBS) was introduced[3,5].Blood spots are collected on a DBS card, and once dried, are not considered hazardous and can be stored for up to three months and transported at ambient temperature without loss of the viral genetic material[5,6]. DBS specimens are also thought to underestimate VL results as compared to plasma specimens. This may result in wrong clinical decisions, especially if the VL is around a threshold of a 1000 copies per ml[7].To compensate for the fact that DBS contains whole blood instead of plasma, with approximately 50% red blood cells, various VL testing platforms recommend a correction factor to make the results comparable with plasma results[7].This is especially important among specimens from PLHIV who may have high prevalence of anaemia [4].”

Line 76-81 reproduced below

“To improve the sensitivity, further research and development has been recommended which includes applying better correction factors to account for haematocrit[7].Previous studies on validity of DBS VL using NASBA either did not clarify whether a correction factor was used or whether a correction factor based on mean haematocrit among PLHIV was applied or did not compare with plasma VL (or any other standard) against a threshold of 1000 copies per ml[9].”

In our first revision, for the sub-group analysis (plasma VL<1000), we found the ICC to be zero (with or without correction). Therefore, for those with plasma VL less than 1000, we interpreted that correlation of DBS with plasma VL results was poor, irrespective of haematocrit correction.
However while analyzing the ICC, we now realise that there was a negative average covariance among items. This violated reliability model assumption (footnote of Table 1). Hence, we have removed the following line from result narrative

“For those with plasma VL less than 1000, the correlation of DBS with plasma VL results was poor, irrespective of haematocrit correction.”

We have discussed the same findings in lines 268-74 of revised manuscript with track changes (reproduced below)

“Our findings confirm that on addition of a haematocrit correction factor, the correlation with plasma VL improved. The type of correction could be based on either an individual or a mean haematocrit corrected factor, depending on the availability of resources. However, among those with plasma VL<1000, we are not able to confidently recommend the use of DBS to assess relative change of VL over time (say from 400 to 800) which is also an early indicator of treatment failure [18,19]. This was due to negative average covariance in the subgroup, violating reliability model assumptions.”

       1.   Thank you for the comment. We have added the data in Table 1. We meant ‘correlation’ and not ‘reliability’. Apologies for this. We have made this edit.  We have described in our methods section as to what we mean by poor, moderate, good correlation (based on ICC and its 0.95 CI).

REVIEWER

       2.   Thank for this comment in round 1 as well as round 2 of review. Area under curve for uncorrected DBS (purple line) - 0.933; mean haematocrit corrected DBS (red line)- 0.933.  Hence, one of the two will get overlapped under the other and this is unpreventable. Hence, we have revised the footnote as “The ROC curve for uncorrected DBS (purple line) is not seen as it is hidden below and overlapped by the ROC curve for mean corrected DBS (red line), their area under the curve are the equal”. We hope it is fine now.

       3a,b. Thank you for your suggestion on the paper by Evans C et al. We have included this and mentioned that it might be important to take patients’ physical conditions into consideration as well. (lines 257-59 of revised manuscript with track changes)

The authors are aware of technical and scientific reasons for lower sensitivity of the DBS  methods, They include

i.DBS sample contain whole blood instead of plasma with approximate 50% red cell.(ref7)

ii There is poor elution of the lysed blood from the filter paper. This can be improved by either increase the elution time or improving the quality of the DBS paper

iii .The poor quality of the DBS specimen prepared by non-laboratory staff .This may be checked by laboratory staff before processing

The low sensitivity (67%) of DBS  in this study is attributed to the nature of the sample.The bulk of the samples have a result below 500 copies(%) and target not detected (45.5%).The sensitivity and reproducibility of DBS results at values below 1000 copies have been demonstrated to be poor

3c.  Pannus etal (ref 7),in their data analysis reported that the bioMerieux recommended the use of 2.29 correction factor for comparison with plasma. We have quoted the correction factor used by the author in the paper.

4.    We have taken this recommendation seriously. Two authors again went through the paper. We then shared the paper with a native English speaker for corrections in English. I hope it is fine now.

5.   There was no difference in mean and individual hematocrit correction methods when analyzing the subgroup DBS (500-999). Among those with DBS(500-999), (n=30), 19 had plasma VL 1000. We have added this in lines 177-79 of revised manuscript with track changes.

“Of 235 with undetectable DBS VL, 22 (9.4%) had detectable plasma VL. A total of 30 (5.8%) DBS specimens had VL between 500 and 999..  Of these 30, nineteen had plasma VL<1000.”
       6.     This correction factor of 2.29 is quoted from the paper by Pannus et al (ref 7). We looked elsewhere including the manuals of the machine provided to us, there is no mention of this correction factor.

       7.   The WHO technical guidelines(84%) sensitivity on DBS are silent on the use of any correction factor. We looked for the reference for this. The report cited an unpublished report which we are not able to access.

Please refer to this link of the WHO report https://apps.who.int/iris/handle/10665/128121. Go to Table 1 of page 10. The reference for 84% is “Lara Vojnov et al., Clinton Health Access Initiative, unpublished manuscript on dried blood spot specimens with highly variable performance compared with plasma across the five available viral load technologies, 15 June 2014”. As shared before, we are not able to access this report.

An important limitation of the evidence supporting DBS specimens for viral load testing is that the majority of studies included in the review used venous whole blood specimens prepared in the laboratory using precision pipettes to dispense the blood onto the filter paper rather than based on specimens obtained in clinical settings. In addition, when plasma specimens are used for viral load monitoring, lack of integrity of the cold chain may influence accuracy. Research is needed to validate the performance of DBS specimens in routine programme settings, with an emphasis on specimen preparation done by less skilled health staff and using different types of filter paper. This paper addressed this point.

Round  3

Reviewer 1 Report

I am deeply worried about this manuscript. It does not disclose any real data but only the analysis results. Thus, readers will not have a mean to check even the existence of the data suggested by the analyses. The authors have not responded to my repeated requests to show some data. Based on my good will as a scientist, I should recommend a rejection here. However, I would like to leave comments to below, in the hope that the authors could reconsider my suggestions and scientifically validate their results in the future.

1. The most essential part of the study is the data and not the analysis results. Without any data, the analysis results could be seen as a fake. Please consider this seriously.

2. If the authors have the data, the paper does not need to depend on the software provided by the company. They should seek for more scientific methods to present their own data.

3. The changing analysis results in the revision process without showing any real data have strongly warned me of the quality of the manuscript. The authors should ask a professional statistician to validate both the data and the analysis results.

4. If the authors are not sure of the correction factor 2.29, they should contact the authors of the publication as well as the company to check it. It is easy and mandatory. Following that, the authors can apply the correction factor to their own data to compare the results with the hematocrit based correction results. Which is better?

5. As well as the authors compare their results with the sensitivity suggested by WHO, they are responsible for the comparison. I mean, it is mandatory that the authors clarify the meaning of the suggested sensitivity (84%). They should contact Dr. Lara Vojnov to discuss and fully clarify the point.

Finally, I wish the authors good luck for the future of the study.